# Metabolomics Intervention towards Better Understanding of Plant Traits

**DOI:** 10.3390/cells10020346

**Published:** 2021-02-07

**Authors:** Vinay Sharma, Prateek Gupta, Priscilla Kagolla, Bhagyashree Hangargi, Akash Veershetty, Devade Pandurang Ramrao, Srinivas Suresh, Rahul Narasanna, Gajanana R. Naik, Anirudh Kumar, Baozhu Guo, Weijian Zhuang, Rajeev K. Varshney, Manish K. Pandey, Rakesh Kumar

**Affiliations:** 1International Crops Research Institute for the Semi-Arid Tropics (ICRISAT), Hyderabad 502324, India; s.vinay@cgiar.org (V.S.); r.k.varshney@cgiar.org (R.K.V.); 2Department of Genetics, Hebrew University of Jerusalem, Jerusalem 91904, Israel; prateekg04@gmail.com; 3Department of Life Sciences, Central University of Karnataka, Kadaganchi 585367, India; kagollap98@gmail.com (P.K.); sharankumarsalagare46362@gmail.com (S.); hangargibhagya@gmail.com (B.H.); akashnaara112@gmail.com (A.V.); sonudevde1999@gmail.com (D.P.R.); srinivassuryavanshi143@gmail.com (S.S.); rahulkambar10@gmail.com (R.N.); grnaik2009@gmail.com (G.R.N.); 4Department of Botany, Indira Gandhi National Tribal University (IGNTU), Amarkantak 484886, India; anirudh.kumar@igntu.ac.in; 5Crop Protection and Management Research Unit, United State Department of Agriculture-Agriculture Research Service (USDA-ARS), Tifton, GA 31793, USA; baozhu.guo@usda.gov; 6College of Plant Protection, Fujian Agriculture and Forestry University, Fujian 350002, Fuzhou, China; weijianz1@163.com; 7State Agricultural Biotechnology Centre, Centre for Crop Research Innovation, Murdoch University, Murdoch, WA 6150, Australia

**Keywords:** metabolome, omics, engineering traits, mQTLs, mGWAS, metabolic engineering

## Abstract

The majority of the most economically important plant and crop species are enriched with the availability of high-quality reference genome sequences forming the basis of gene discovery which control the important biochemical pathways. The transcriptomics and proteomics resources have also been made available for many of these plant species that intensify the understanding at expression levels. However, still we lack integrated studies spanning genomics–transcriptomics–proteomics, connected to metabolomics, the most complicated phase in phenotype expression. Nevertheless, for the past few decades, emphasis has been more on metabolome which plays a crucial role in defining the phenotype (trait) during crop improvement. The emergence of modern high throughput metabolome analyzing platforms have accelerated the discovery of a wide variety of biochemical types of metabolites and new pathways, also helped in improving the understanding of known existing pathways. Pinpointing the causal gene(s) and elucidation of metabolic pathways are very important for development of improved lines with high precision in crop breeding. Along with other-omics sciences, metabolomics studies have helped in characterization and annotation of a new gene(s) function. Hereby, we summarize several areas in the field of crop development where metabolomics studies have made its remarkable impact. We also assess the recent research on metabolomics, together with other omics, contributing toward genetic engineering to target traits and key pathway(s).

## 1. Introduction

Metabolomics in the plant system has extended the opportunities towards the discovery of new pathways and integrating it with other omics-based data generated from genomics, transcriptomics, and proteomics, which improved existing genome annotations. The study of metabolomics has gained attention in the last 20 years, as most of the research labs were involved in generating the metabolic profile through various platforms such as nuclear magnetic resonance (NMR), liquid chromatography-mass spectrometry (LC-MS), and gas chromatography–mass spectrometry (GC-MS), which also lead to enrichment of several metabolite databases such as KEGG, GOLM, NIEST databases. By 2010, most of the metabolomics labs were equipped with the latest analytical high-throughput chromatography instruments. It is coupled with highly sensitive and precise mass spectrometric tools developed through revolutionary advances in the field of mass-spectrometry and data processing softwares including the free web-tool like Metaboanalyst and offline software METLIN. The most important plant-based metabolite data-processing tools involves platforms such as ChromaTOF, Met-Align, MET-COFEA, MET-XAlign, etc. [1]. Further, availability of statistical tools, such as MetaboAnalyst, Cytoscape, Statistical analysis tool, etc., have made statistical analysis simple, such as principal component analysis (PCA), partial least squares (PLS), K-means clustering, boxplot, heatmap, and reconstructing metabolic pathways [1,2,3]. The availability of the above tools has allowed analysis of a remarkable collection of metabolome data from the samples that were extracted for the analysis of primary and secondary metabolites, and lipidomics under various growth conditions. Metabolome data are available for several model and crop species including *Arabidopsis thaliana*, *Arachis hypogaea*, *Actinidia Lindl*. spp., *Citrus* spp., *Lotus* sp., *Lupinus albus*, *Helianthus annuus* L., *Mangifera indica*, *Medicago trancatula*, *Malus* spp., *Fragaria* × *ananassa*, *Glycine max*, *Oryza sativa*, *Pyrus communis*, *Solanum lycopersicum* L., *Vitis vinifera*, *Zea mays*, etc., [1,4]. The metabolomics study was done to explore multiple areas such as biotic stress [1,5,6], abiotic stress [7,8,9], legumes and cereals quality improvement [10,11,12,13,14,15,16,17], biofuel production and lipid profiling [18,19,20,21], impact of climate change and high CO_2_ level [22,23,24,25], hormone profiling [26], and improving fruit quality [1,26,27,28,29]. These attempts have provided opportunities to dissect the metabolic pathways for developing stress-tolerant and nutrition-rich crop plants [1]. Previously, several review articles have focused on providing the detailed methodology and availability of the advanced instruments which are being used for the omics study including metabolomics [1,30,31]. In this review, we have covered the important area that has flourished in the era of metabolomics and how the knowledge gathered through metabolomics has helped in dissecting different pathways through metabolic engineering for crop improvement.

## 2. Integrating Metabolomics with Genomics Study for Gene Characterization and Metabolomics-Assisted Breeding

Over the past decade, metabolomics has seen excellent progress in the area of development of instrumentation and software advancement; providing the opportunity to analyze the whole metabolome of plant species using high throughput methods. Metabolomics applications have supported several research areas, especially biotechnology, genomics, molecular plant breeding, and functional genomics [32]. In addition, its use makes advances in the area of translation metabolomics and plant breeding. Recent advancements in post-genomics technologies have boosted the process of screening and metabolomics integrations with other high throughput methodologies, which will be reducing the time required to develop crop varieties with enhanced biotic and abiotic stress tolerance. Metabolomics has a strong ability to holistically explore the evaluation and phenotyping of various metabolites in crops [33]. Approximately 840 metabolites were identified in rice cultivars that could be used in breeding programmes [34]. mQTLs (metabolomic quantitative trait loci) mapping and mGWAS (metabolic genome-wide association studies) are important approaches for the identification of genetic variants associated with metabolic-related traits [10].

### 2.1. Metabolomic Quantitative Trait Loci

To understand the metabolic networks that regulate the complex developmental process metabolomics-based quantitative trait locus (mQTL) studies are important for improving the quality and performance of elite cultivars. In addition, results obtained from mQTL studies contribute to a deeper understanding of quantitative and functional genetics [35]. Metabolic profiling decreases the gap between phenotype and genotype and offers new opportunities for metabolic dissection, starting with the discovery of molecular markers along with mQTL mapping studies for the identification of candidate genes and linked genomic region. Metabolic markers have become an important tool to uncover and investigate the various biological complex pathways responsible for distinct phenotypes [36]. The mQTL approach connects the metabolome and genome, and provides important insight into genetic function and investigates phenotypic variation via metabolic profiling and comprehensive gene expression analysis [37].

Advances in genomic technologies have enabled mQTL detections via high-density maps for candidate gene discovery [38]. Several candidate genes that regulate metabolites biosynthesis have been detected using multi-omics approaches with reverse and forward genetics methods [39]. Moreover, population genetics, which integrates quantitative genetics with metabolic profiling has begun to explore genetic regulation of the entire metabolome in plants. A recent study, reported by [10], uses high-density map with 1619 bins for mQTL mapping, leading to identification of several mQTLs for flag leaf and germinating seeds across 12 linkage groups in rice. Comparative mQTL studies in two rice cultivars showed tissue-specific secondary metabolites accumulation under strict genetic regulation. A total of 19 metabolites have been identified on 23 mQTLs, indicating a substantial interaction between metabolites and the associated genomic loci [10]. Another mQTL study conducted in back-crossed inbred lines (BILs) of rice identified 700 different metabolic characteristics under 802 mQTLs which show an unusual range that could regulate various metabolic traits [40]. Further, in maize, 26 distinct metabolites were identified which shows a strong association with single nucleotide polymorphism (SNPs), and highlighted the importance of cinnamoyl-CoA reductase gene located on chromosome 9 for controlling lignocellulosic biomass [41].

mQTL mapping is an effective method for identifying stress-responsive trait pathways. In the barley recombinant inbred line (RIL) population, the mQTL study detected 98 different stress-responsive metabolites and observed that their abundance modulates through a coordinated expression of several genes to function under drought conditions [42]. Similarly, the mQTL study in barley identified 57 metabolites under drought stress conditions [43]. In rapeseed, metabolic profiling and gene function analysis to identify the basis of glucosinolate synthesis was performed, which reported around 105 mQTLs in seeds and leaves involved with glucosinolate production [44]. In a very recent study carried out in the tomato wild and introgression lines, 679 mQTLs were identified for secondary metabolism-related pathways linked to environmental stress tolerance [45]. In later experiments, mQTL analysis was performed in a similar IL to investigate metabolite concentration [46]. Likewise, metabolic profiling of wheat (double haploid lines) by LC/MS method revealed about 558 secondary metabolites, comprising alkaloids, flavonoids, and phenylpropanoids [47]. The GC-TOF/MS-based metabolic analysis of seed of tomato RIL population was performed to investigate the seed metabolism [48], which identified several genomic regions controlling a group of metabolites. As sequencing technologies progresses, more plant genomes have been sequenced and these high-quality genomes may further accelerate the crop plant’s mQTL studies, leading to establishing a relationship between genome and trait expression. For example, phenylpropanoid synthesis genes have been identified in corn [49], phenolamide in corn and rice [50,51], and glucosinolate regulation in cabbage [52] have been reported; these by-products are regarded as defense responsive metabolites. In the future, these mQTLs will help in targeting several pathways for designing crops with desired traits.

### 2.2. Metabolic Genome-Wide Association Studies

The mGWAS was developed as a valuable tool to explain the natural genetic basis of different metabolic shifts in a plant’s metabolome (Table 1). Recent studies have shown the broad perspective of plant metabolites related to specific traits [16]. A parallel study of mGWAS with phenotypic genome-wide association studies (pGWAS) in rice have effectively detected novel candidate genes that control the genetic variation in relevant agronomic traits [16]. Metabolic polymorphism studies in rice species reported various forms of flavone glycosylation and stated a positive association between plant growth conditions and UVB light exposure [53]. A recent mGWAS study in rice reported 323 associations among 89 secondary metabolites for two genetic architecture types, related to secondary metabolite concentration [54]. Natural variation studies and the metabolic profiling of phenolamides have been undertaken by Dong and colleagues using an LC/MS mediated targeted metabolomics method in several rice accessions. They identified a temporal and spatial accumulation of several phenolamides. In addition, mGWAS detected two spermidine hydroxyl cinnamoyl transferases, responsible for natural variations in spermidine levels. This study showed that gene-to-metabolic analysis through mGWAS offers an opportunity to improve crop genetics [51]. Another mGWAS study was conducted to analyze rice metabolism biochemical and genetic variants. The study reported 36 genes linked to specific metabolites that regulate physiological and nutritional-related traits [34]. Traits associated with primary and secondary metabolites could be utilized as metabolic markers to promote plant breeding. Similarly, the maize mGWAS study was conducted to reveal complex metabolic character. Around 26 metabolites associated with SNPs have been detected which regulate the main target of cinnamoyl-CoA reductase to increase the lignocellulosic quality of maize [41]. Recently, in winter, wheat metabolic profiling has been done to make apparent the association of 18,372 SNPs and detected 76 metabolites. The relation between metabolites has shown a functional relationship with several pathways of the Krebs cycle. The mGWAS identified a strong correlation between 1 and 17 SNPs with six metabolic attributes. These findings provide a way to predict the impact of genetic interventions on related metabolic traits and possibly, on a metabolic phenotype [55]. These studies will speed up metabolomics-assisted breeding to improve the quality and quantity of target traits in crops.

### 2.3. Metabolic Analysis for Biotic Stress Tolerance in Crop Plants

Recent evidence showed that invasive microbes systematically suppress plant immune function in susceptible cultivars using protein-effector molecules which can also be identified by plant R gene products in inconsistent interactions [61]. Besides counteracting plant defenses, an effective pathogen must also subvert host plant metabolism to facilitate efficient intake, sequestration, and use of host-derived nutrients [62,63]. Several studies have utilized transcriptional profile analysis to examine the global changes in expression of genes which arise during host invasion by biotrophic and hemibiotrophic fungi [64,65,66], and have reported co-ordinated expression of several gene products, that often have a predicted metabolic function. Therefore, a metabolome study related to the stress responses is important to unravel the molecules/metabolites which coordinate susceptibility and/or resistance traits in different plant [1,7,8,9,67,68,69,70,71,72].

Biotic stress resistance-associated loci have been reported in various crop diseases such as late blight of potato (*Phytophthora infestans*) [73], rice blast (*Magnaporthe grisea*) [74], and cereal rusts (*Puccinia* spp.) [75]. Two mQTLs, *Qfhs.ndsu-3BS* in barley [76] and *Fhb1* in wheat, have been also reported for Fusarium head blight disease resistance [77]. Such loci generally co-localize multiple genes and cloning of such loci to identify all the co-localizing genes is a challenging task. A combined transcriptomics and metabolomics analysis of the rice in response to bacterial blight pathogen *Xanthomonas oryzae* pv. *Oryzae* reported that few mRNA and metabolite differences have been observed, and many differential changes in the *Xa21*-mediated response occurred [78]. Important transcriptional induction of various pathogenesis-related genes in the *Xa21* challenged strain, as well as differential expression of *GAD*, *PAL*, *ICL1*, and *Glutathione-S-transferase* transcripts suggested a minimal association with changes in metabolite under single time point global profiling conditions. In fact, a metabolome study using LC-MS and GC-MS methods identified several hundreds of compounds, which were modulated when the susceptible and resistant line was compared. Most importantly, this study identified ornithine, citrulline, tyrosine, phenylalanine, lysine, oxoproline, butyrolactam, and N-acetylglutamate as the key compounds involved in providing tolerance against bacterial blight pathogen in rice. Additionally, the role of acetophenone and 2-phenylpropanol (acetophenone reduction product) was identified during host resistance, as earlier these were reported to be involved in the dicot plants [79]. More importantly, recently through metabolomics study, resveratrol was identified to have inhibitory action on *Xoo* as it causes oxidative stress as well as disrupts several pathways related to *Xoo* growth and metabolism including amino acid, purine, energy, and NAD^+^ metabolism in *Xoo* [80]. Further, metabolomics was deployed for the reconstruction of a genome-wide metabolic model of *Xoo* and revealed the influence of nitrogen-fertilizers on *Xanthomonas oryzae* pv. *Oryzae* metabolism, a differential flux in nitrogen-metabolism and ammonia uptake was observed [81]. Like bacterial blight, Asian rice gall midge (*Orseolia oryzae*) is a severe rice pest causing major yield losses. Metabolic studies reported a number of metabolites that can be categorized as resistance, susceptibility, infestation, and host features, depending on their relative occurrence, and can be considered as biomarkers for insect–plant interaction in general and rice–gall midge interaction in particular [82]. Therefore, more metabolomics studies including tissue and single cell-specific studies are required to develop interactome maps by integrating different layers of omics studies.

## 3. Important Achievements through Metabolic Engineering

In the past two decades, several attempts have been made towards characterization of genes related to important metabolic pathways which have also led to the improvement of several crop plants in the area of bio-fortification. We have summarized most of them in Table 2 and discussed some important ones below.

### 3.1. Fortification of Carotenoids and Flavonoids

The carotenoid biosynthesis and metabolism are studied intensively as different carotenoids have distinct nutraceutical roles such as lycopene as an antioxidant, lutein for vision, acyclic carotenoids i.e., phytoene and phytofluene in nutricosmetics, and β-carotene as the primary dietary precursor of vitamin A. The sufficient intake of vitamin A is essential for human health. In many developing and under developed countries, vitamin A deficiency (VAD) is a prevalent cause of premature death and childhood blindness. In addition, therapeutic doses of β-carotene have protective effects against cardiovascular disease, certain cancers, and aging-related diseases [166,167]. Considering the nutritional benefit of β-carotene, in recent years, considerable efforts have been directed to elevate its content in food crops. Various metabolic engineering approaches have been used to increase the β-carotene levels to alleviate the provitamin A deficiency, beginning from “Golden Rice I”. Since then, biofortification is attempted in several crop plants using transgenic approaches, conventional breeding, and screening genetic diversity. Conventional breeding and marker-assisted selection have significantly increased carotenoid content in a few instances, but there is the need for identification of novel alleles or wild germplasm associated with high carotene levels [168,169,170]. On the other hand, transgenic approaches using overexpression of plant genes or introduction of bacterial genes lead to high provitamin A, but suffer from GM regulations, safety, and public acceptance [124,171,172,173]. Screening of natural accessions, genetic variants, and mutants with altered carotenoid content provides a faster and safer way for the biofortification of provitamin A in crop plants [174,175]. Carotenoid sequestration was also achieved via overexpression of *Orange* (*Or*) gene or *Or* mutants harboring “Golden SNP”, which encodes the plastid-localized DnaJ cysteine-rich protein, has been successfully demonstrated in melons, cauliflower, and potato tubers [176,177]. A list of provitamin A biofortified crops is summarized in Table 3. Not only provitamin-A carotenoids, but xanthophylls like zeaxanthin and lutein also play an imperative role in protection against age-related macular degeneration (AMD) which is the predominant cause of blindness in several countries [178,179]. Recently, a zeaxanthin-rich tomato fruit was developed using metabolic engineering and genetic breeding which has highest concentration of zeaxanthin achieved in a primary crop [180]. To date, the exploitation of several natural and transgenic resources has been utilized for the biofortification of carotenoids in crop plants and the field is still expanding by identifying new regulatory factors which can modulate the carotenoid production.

Flavonoids, belong to a group of polyphenolic plant secondary metabolites, which not only have physiological roles in plants but also constitute our daily diet. There are six major subclasses of flavonoids notably, anthocyanidins, flavan-3-ols, flavonols, flavanones, flavones, and isoflavones, which are widely present in fruits and vegetables. Flavonoids-rich fruits and vegetables have been largely promoted in the human diet because of their broad spectrum of health-promoting benefits, which include anti-oxidant and anti-inflammatory properties. Given its nutritional importance, several efforts have been made to increase flavonoid levels in various crops using overexpression of key structural genes and transcription factors. Overexpression of single or multiple structural genes from different sources resulted in a significant increase in flavonoid production. Schijlen et al. [202] showed that combining structural flavonoid genes *stilbene synthase*, *chalcone synthase*, *chalcone reductase*, *chalcone isomerase*, and *flavone synthase* lead to the accumulation of stilbenes, deoxy chalcone, flavones, and flavanols in tomato peel. Similarly, overexpression of *petunia chalcone isomerase* in tomato fruits resulted in increased flavanols levels [203]. In addition, several transcription factors have been used to regulate phenylpropanoid metabolism. Bovy et al. [204] utilize maize transcription factor genes *LC* and *C1* for production of high flavanols tomato. Likewise, Zhang et al. [205] reported fruit-specific expression of *AtMYB12* in tomato leads to the accumulation of flavanols. Accumulation of anthocyanins in tomato fruits was achieved by expressing snapdragon transcription factors *AmDel* and *AmRos1* [206]. Recently, Jian et al. [207] showed the overexpression of *SlMYB75* promotes anthocyanin and flavonoids accumulation. These results suggest that structural genes and transcription factors together can be used to achieve a higher accumulation of flavonoids in crop plants.

### 3.2. Metabolic Engineering of Phytohormone Signaling and Biosynthetic Pathway to Improve Crop Performance

Phytohormones auxins, brassinosteroids (BRs), cytokinins (CKs), ethylene, gibberellins (GAs), and abscisic acid (ABA) are the key regulator of the plant architecture and their growth [26,208]. In the recent past two decades, several transgenics have been generated to understand their role and also to improve the crop plants [26]. In fact, one of the most key events in plant biology and agronomy was that the selection of the semi draft variety in wheat and rice during the green revolution was driven by a selection of genes related to GA pathways such as *GA-20 oxidase* and *Della* [209,210]. One of the key transcription factors regulating GA signaling is *Squamosa promoter-binding-like protein 8* (*SPL8*), amputation, or attenuation of it through transgenic approach severe declines GA accumulation via *GA2-OX* and *GA2-OX6* [211]. Likewise, cytokinin biosynthesis was targeted to alter plant architecture, growth habit, and life cycle because upregulation of cytokinin production enhances biomass and delays plant senescence via cell division [212]. A mutation in the cytokinin receptor or overexpression of gene *cytokinin oxidase* (*CKX*, encode for cytokinin catabolizing enzyme) can lead to the smaller shoot apical meristem, decreased leaf area, and severely retard plant growth [213]. Therefore, to achieve better crop yield, *CKX* gene homologs were targeted by developing knockouts. In rice, *CKX* knockout results in the improved maintenance of photosynthetic rate, panicle branching, and reduced yield gap under salinity stress [214]. Several attempts involved upregulation of cytokinin through overexpression of a cytokinin biosynthetic genes *isopentenyl transferase* (*ipt*) in broad bean [215], creeping bentgrass [216], peanut [217], rice [218], tobacco [219], and in salinity stress exposed cotton [220]. Additionally, transgenic poplar plants overexpressing a *YUCCA6*, abiotic stress-responsive gene involving in tryptophan-dependent IAA biogenesis pathway, exhibit remarkable rapid shoot elongation with restricted tap root but with enhanced root hairs [221].

The complete knowledge of metabolic pathways is very important. Recently, a cluster of genes related to ABA signaling was targeted through genome editing to improve drought tolerance, due to which the edited lines showed a remarkable 30 percent yield increase due to increased number of spikelet numbers per main panicle [222]. The edited genes involved ABA receptor (*RCAR*) family of proteins *PYL1*–*PYL6*, *PYL12*, *PYL7*–*PYL11*, and *PYL13*. ABA plays a key role in abiotic stress tolerance especially during drought stress, as a result, several ABA signaling and biosynthetic genes including *ABA-responsive complex* (*ABRC1*) and *9-cis-epoxy carotenoid dioxygenase* (*NCED*) have been targeted to improve the abiotic stress tolerance in crop plants [223,224]. Lee et al. [223] demonstrated the role of *ABRC1* in tomato transgenic in maintaining yield against cold, drought, and salinity stress. Likewise, the gene *NCED1* was overexpressed in tobacco to achieve tolerance to drought and salt stress due to enhanced accumulation of ABA in leaves [224].

### 3.3. Engineering of Cell Wall Biosynthesis Pathway: Some Examples

The non-living cell wall present in the plant system makes them unique compared to animal cells, provides structural and mechanical support to the whole cell, and also acts as a physical barrier against both abiotic and biotic stresses. The principal compositions of a cell wall are cellulose, hemicelluloses, and lignins. Often, the plant activates the cell wall metabolism-related pathways whenever they are challenged with stress, such as higher production of lignin biosynthesis enzymes during biotic and abiotic stresses. Therefore, immense progress has been made to target cell wall-related pathways to confer tolerance against these biotic and abiotic stresses. Modification of the lignin biosynthetic pathway was done in *Pinus radiate*, which provided the significance of gene *4-coumarate–Co A ligase* in the accumulation and distribution of lignin in the tracheid element during cell wall and wood formation; by which it also interferes into plant height [225]; indicating its economic importance in the field of horticulture for generating a dwarfed plant or “bonsai tree-like”. The biosynthesis of the cell required UDP-Glc, which is required for the formation of different sugars required during wall formation [225]. Researchers have explored genes *UDP-glucose pyrophosphorylase* and *sucrose synthase* for drought tolerance as their overexpression causes enhanced cellulose accumulation by increased production of UDP-Glc [226]. Likewise, the role of the cellulose biosynthetic gene *cellulose synthase* was observed in *Brassinosteroid insensitive2* mutants [227]. Further, the *Expansin* gene, which controls cell wall loosening, plays a very important in the root architecture during drought tolerance [228]. The gene *SHINE* encodes the AP2/ERF transcription factor family protein known to control the wax biosynthesis pathway in a plant [229]. In rice, the gene *SHINE* was overexpressed, which led to reduced 45% lignin content and increased cellulose content by 34%, thus improving the fodder quality and digestibility [230]. The silencing of the *NAC2* transcription factor, which binds to the promoter region of *Expansin-A4* (*EXP-A4*), caused reduced drought tolerance during floral organ development in rose due to reduced expression of gene *EXP-A4* [231]. On the contrary, overexpression of *EXP-A4* in Arabidopsis showed an expected drought tolerance phenotype [231]. In rice, overexpression of *Sucrose synthase* (*SUS*) led to increased cell wall-related polysaccharides deposition and reduced cellulose-crystallinity as well as xylose/arabinose proportion in hemicellulose; which is beneficial for the biofuel industry [232]. The genetic engineering of the cell wall biosynthetic pathway through overexpression of *SUS* in rice added a new dimension towards its role in the cell wall metabolism.

### 3.4. Metabolic Engineering for Bio-Fortification of Phytonutrients

In the past 20 years, several attempts have been made to enrich the nutritional constitution in crop plants; so that they can emerge as a superfood; such as development of the purple tomato [206], where a gene was overexpressed for a hyperaccumulation of “anthocyanin” which is an anticancerous compound. One of the most important contributions in the field of metabolic engineering of crop plants was the development of ‘Gloden rice’ by overexpressing *phytoene synthase* (*PSY*) from maize and the daffodil plant, and *PSY* ortholog from (*Erwinia uridovora*) bacterial using the endosperm specific promoter, leading to a 27-fold increase in the β-carotene level in the transgenic golden rice [1,124,171]. Every year, folate deficiency causes death, cardiovascular disease, megaloblastic anemia, and neurological disorder in newborns [1]. Now, due to the characterization of the folate biosynthesis pathways genes, several genes have been overexpressed in Arabidopsis, lettuce, tomato, lettuce, maize, and potato [1]. The gene *GTP-cyclohydrolase 1* (*GTPCH1*) was overexpressed in Arabidopsis, lettuce, rice, and tomato [233,234,235,236].

## 4. Study of Root Nodule Symbiosis (RNS) in Legumes

The symbiotic nitrogen fixation is mainly restricted to legumes, there are several rhizobia including certain diazotrophs that inhabit the rhizosphere of other crops, which are involved in plant development. In the late 19th century, legumes (Fabaceae) were found to be capable of forming a root nodule symbiosis (RNS) with nitrogen-fixing rhizobia which improves soil fertility [237]. With the emergence of modern tools such as transcriptomics and proteomics, the molecular mechanism of root nodule symbiosis (RNS), nodule organogenesis, and their development have been well studied in model legume species [238,239]. These studies have centered the concepts that mark the path for the engineering of nitrogen fixation nodule symbiosis which include; various blueprints for nitrogen-fixing root nodule symbiosis (RNS), use of non-model crops to recognize important symbiosis genes, recruitment of the arbuscular mycorrhizal pathway for RNS, and crosstalk between developmental programs involved in plants and RNS. Not only do these concepts reflect significant breakthroughs in our knowledge of RNS, but they also provide important insights for engineering strategies possibilities and constraints. Various studies in legumes reported a number of genes which are associated with RNS (Figure 1) [240,241,242,243,244,245]. Some important genes which control the RNS have been reported: *NFR*, *LYK3*, *LYR3*, *DMI1-3*, *CASTOR*, *POLLUX*, *NIP85*, *NUP133*, *NENA*, and *SyMRK* Nod factor for perception, and the downstream signaling pathway includes transcription factors NSP1, NSP2, ERN1, etc. (See Figure 1) [238,239]. More such studies are required in order to understand the molecular biology, biochemistry, and nodulation physiology in nodulating species.

## 5. Addressing Symbiotic Nitrogen Fixation in Cereals and Non-Legume Crop Plants

The nitrogen-fixing orders Cucurbitales, Fagales, Fabales, Rosales, and other Poaceae (Poales) varied widely and their root systems showed various developmental adaptations [246]. The crop plants such as cereals demand a significant amount of nitrogen for their proper growth and grain production, therefore engineering of these crops would be ideal to induce nitrogen fixation nodulation-related traits [247]. Selection of a single gene for metabolic engineering of non-legumes plants (such as cereals) to induce root nodulation for better nitrogen use efficiency is the biggest challenge. Therefore, by comparing the various RNS and the associated genes, we can distinguish common features and the core genes that must be recruited in the early development of the trait. However, knowledge and understanding of these genes can also be important, as they can be related to processes like root hair invasion, nodule organogenesis, and symbiosome development, thereby enabling an engineering approach that integrates features from multiple symbioses. In order to assess a core community of symbiosis genes important to RNS and to classify lineage-specific adaptations, it is necessary to choose representative species in different clades for comparative study. Particularly the latter is a pro, as CRISPR-Cas9-based reverse genetics will allow the study of the function of genes.

Introducing a cluster of genes responsible for the root nodulation through genetic engineering will be an important achievement; in fact, such novel attempts are required in cereals and other non-legume crops [248,249,250,251]. If all genes in model species are defined for nitrogen-fixing symbiosis, it will provide a framework for engineering in far-related species. Since the nitrogen-fixing trait is believed to have a single evolutionary origin, several species in nitrogen-fixing clade may lose nodulation in the future [252,253]. A current approach is to bring back mutated genes of symbiotic association (nitrogen-fixing clade) in non-nodulating species. Likewise, the species representing a sister lineage of a clade could be approached [252,254]. In non-nodulating species, introduction of nodulation will rely on the endogenous genes, but several transgenes are required to transfer. At first, *NFP*/*NFR5*/*NFP2*, *NIN*, and *RPG* genes can be used. The question still stands whether these genes are the only genes that are responsible for nodulation [255]. Other genes such as leghemoglobin encoding have most likely undergone minor but important adaptations [256].

Expecting functional RNS in a single attempt in non-nodulating species is not possible as it is coordinated through multiple genes. Instead, engineering might be an iterative approach. Evolutionary genomics studies indicate that relatively few genetic elements are required to provide nitrogen-fixing ability from legume to non-legume species [257]. The transfer of nitrogenase encoding genes to plants needs a bacterial concatemerization genetic unit (a minimum set of three genes) [258]. Engineering nitrogenase encoding bacterial *nif* genes into non-legumes species is quite difficult because of the complex nature of nitrogenase biogenesis and nitrogenase sensitivity in the presence of oxygen. Advanced genetic and biochemical studies have defined the common core group of genes that are needed for the functional biogenesis of nitrogenase [259]. Moreover, potential low-oxygen subcellular conditions provided by mitochondria and plastids to express active nitrogenase activity in plants enable this engineering approach [260]. Recent studies have shown that the legume symbiotic signaling pathway (SYM) plays a key role in arbuscular mycorrhizal symbiotic associations (AMSA). Various plants including cereals could form AMSA, but they do not have the ability to form nitrogen-fixing nodules. The SYM pathway for the arbuscular mycorrhizal associations in cereals can be engineered to perceive rhizobial signal molecules, which can trigger this pathway and activation into an oxygen-limited nodule-like-root organ for fixation of nitrogen [261]. Prior phylogenomic studies have shown that a set of genes can convert a species in AMSA into a nitrogen fixation symbiosis [252,256]. In cereals, chloroplasts and mitochondria are known to be ideal locations for generating a high-energy nitrogenase enzyme [262]; however, oxygen evolved from chloroplasts during photosynthesis could disrupt the nitrogenase enzyme complex formation. A potential solution is spatio-temporal separation of photosynthesis and nitrogen fixation, which means that *nif* genes could express only in dark periods or in non-photosynthetic parts (root system) [263]. Besides, a carbon-secretion approach that promotes increased carbon competition among the nitrogen-fixing population can be used to develop adequate signals between cereals and nitrogen-fixing rhizobia for effective colonization [261].

Phylogenomics studies assisted *de novo* genome sequencing of non-model legume species led to a better understanding of the origin of nodulation trait. These studies have paved the path for trait engineering. These comparative phylogenomic studies were comprehensive, as result more target genes were being found, that encouraged researchers to put efforts towards the genetic engineering for nitrogen fixation symbiosis-related traits. Metabolic engineering of nitrogen fixation pathway such as genes associated with N transport, assimilation, and primary N metabolism for the improvement of nitrogen use efficiency (NUE) in crop plants is important and appeared to be most promising [264,265,266,267]. In addition, there are several genes, which are involved in C metabolism, and appeared to have a close connection between C and N metabolism, it is hoped that modification of these genes could improve N uptake [265]. There is an amino acid biogenesis gene, *AlaAT*, which when overexpressed in canola and rice, exhibits an NUE phenotype in the greenhouse and field condition [268,269]. This gene encodes for alanine aminotransferase (AlaAT, EC.2.6.1.2), an enzyme that catalyzes the reversible synthesis of alanine and 2-oxoglutarate from pyruvate and glutamate, resulting in N metabolism downstream of GS and GOGAT pathway. Intriguingly, transcriptomics analysis of *alanine aminotransferase* (AlaAT-ox) over-expressing rice lines with wild type (WT), under low, medium, or high N conditions, did not detect any of the known N transport and N-assimilation genes as differentially regulated, instead, the highly differentiated genes were regulatory transcription factor associated with secondary metabolism, and few genes with unknown function [270,271]. Due to the change in the expression of the TCA and secondary metabolite-associated genes, researcher focused on the assessment of N-containing metabolites and the N-flux balance in transgenic plants [272]. In our view, research efforts in this direction is important, because crops engineered for RNS may have a promising future in the incoming era.

## 6. Public Perception for the Metabolic Engineered Plants

In the present world, every year, the food demand is increasing; on the other side, the agriculture system is degrading and arable land is shrinking due to severe thinning of biodiversity and increased incidence of climate change-driven uncertainty in rain. Therefore, in the present scenario, a traditional breeding-based outcome may take reasonable time to fulfill the demand; the breeders must adopt molecular biology as a tool to develop climate smart crops. One of the important achievement in the field of plant biotechnology is development of transgenic tomato “flavor saver” (Flavr Savr or CGN-89564-2), developed by Monsanto [273]. Similar to Flavr Savr, many important crops were developed by targeting metabolic pathways for enhancing the postharvest shelf-life or biotic and abiotic stress tolerance [274]. In plant breeding, genetic engineering has played a very important role, as a result around 525 transgenic events, of which maximum 238 events is registered for maize, 61 for cotton, 49 for potato, 42 for canola, 41 for soybean, etc., and worldwide nearly 32 crops have received approval for cultivation [275]. However, from the past two decades, frequently outrage from the public and NGOs was observed against transgenic and/or genetically modified crops (GMOs) including Flavr Savr which was approved for sale by the Food and Drug Administration (FDA), USA [273]. Now, in the present era, genome/gene(s) editing has made a significant impact; earlier, ZFNs and TALEN played very important roles and the products are already available in the market [274,275,276]; several countries like US, Canada, China, etc. have shown positive response to their product and treated them just as mutants; unlike EU’s regulations which are stringent and treated these genomes edited crops as the transgenic. In July 2018, ECJ (European Court of Justice) stated that “All genome-edited plants should be treated legally as genetically-modified organisms (GMOs), using definitions dating from 2001”. Now, with the advent of the CRISPR/Cas, a revolutionary genome/gene editing tool, the regulatory barrier is expected to get weaken in coming years [274,275,276] as the regulatory agencies of several countries such as USA, Canada, China, etc., have considered them as mutants [276]. In addition, the technique CRISPR/Cas can more favorably modified and used as several variants of Cas enzymes are now available [277]. In the present scenario, CRIPSR/Cas is considered as one of the best tools for editing the traits in crop(s) species. Additionally, technique such as speed breeding can be integrated to achieve more from CRISPR/Cas.

## 7. Future Perspective

In future, the *de novo* domestication would become one of the most important areas. To achieve *de novo* domestication, metabolomics assisted breeding and the knowledge of metabolic pathways will play very important role. Earlier, during ‘Green Revolution’, the selection of genes related to GAs pathways have played a crucial role in the development of semi dwarf high yielding variety, which helped in fulfilling the food demand of billions of people. Today, a better understanding of a metabolic pathway through an integrated approach can redesign the ancestral species, which are resistant to several biotic and abiotic stresses. In addition, the advent of modern sequencing technology has been playing a pivotal role in fine-tuning the genome annotation by utilizing available transcriptome, proteome, and metabolome atlas data. Therefore, utilization of metabolomics data would help in the rapid generation of climate-smart and bio-fortified nutrient-rich varieties to achieve targeted sustainable food production and security.

## Figures and Tables

**Figure 1 cells-10-00346-f001:**
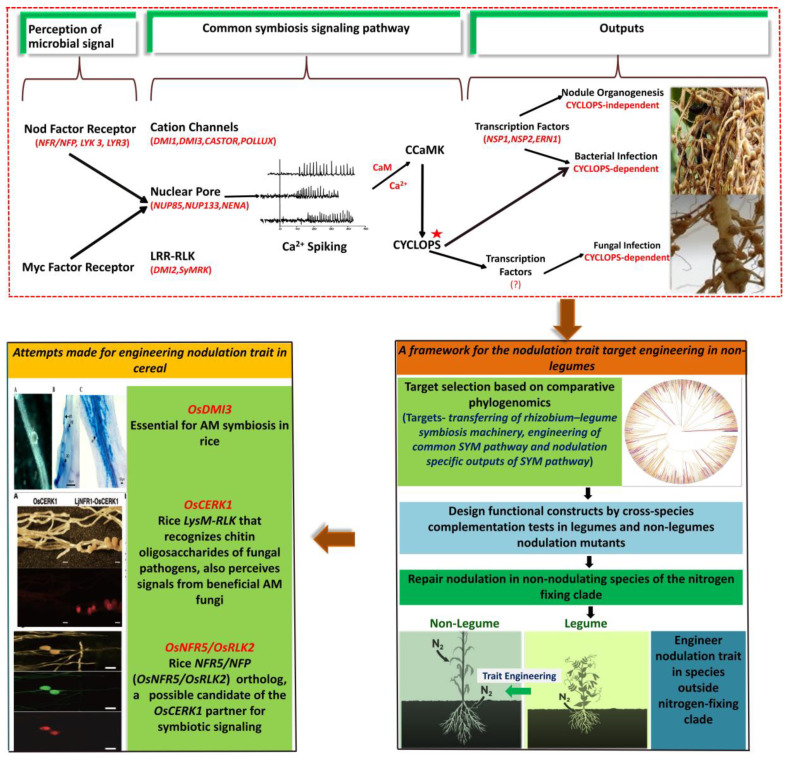
Schematic diagram representing the current advancement, opportunities towards understanding the nitrogen metabolism, root nodulation mechanism, and their implementation in non-legume crop plants.

**Table 1 cells-10-00346-t001:** Metabolomics-assisted breeding studies.

Crop Name	Population	Target Traits	Sample Tissue	Profiling	Significant Outcome	Reference
*Oryza sativa*	Zhenshan 97 × Minghui 63 (RIL)	Metabolome	Flag leaf and seed	Liquid chromatography (LC)–electrospray ionization (ESI)–MS/MS system	Identified twenty-four candidate genes, underlying phenolics, and related pathways	[10]
*Oryza sativa*	Sasanishiki × Habatak (BIL)	Metabolome	Seed	Liquid chromatography-quadrupole-time-of-flight-mass spectrometry	Identified genomic region and genes potentially involved in the biogenesis of apigenin-6,8-di-C-a-L-arabinoside	[40]
*Triticum aestivum*	Excalibur × Kukri (DH)	Metabolome	Flag leaf	Liquid chromatography electrospray ionization tandem mass spectrometric	Identified five major phenology-related loci	[47]
*Triticum aestivum*	KN9204 × J41 (RIL)	Metabolome	Kernel	Liquid chromatography-mass spectrometry	Identified 1005 mQTLs, linked with 24 candidate genes which modulating different metabolite levels, of which two genes are involved in flavonoids synthesis and modification.	[56]
*Zea mays*	BB RIL lines (197) and ZY RIL lines (197)	Metabolome	Mature Kernel	Liquid chromatography-mass spectrometry	Identified candidate genes for maize quality improvement	[37]
*Zea mays*	B73 × By804 (RIL)	Primary metabolism	Leaf at seedling stage, leaf at reproductive stage, and kernel	Gas chromatography time-of-flight mass spectrometry	Identified 297 mQTLs for 79 primary metabolites across three tissues	[35]
*Hordeum vulgare*	Maresi × CamB (RIL)	Metabolome	Flag leaf	Liquid chromatography–mass spectrometry	Reported mQTL in a genomic region of SNP 3011-111 and SSR Bmag0692 have linkages with metabolites	[42]
*Hordeum vulgare*	Landraces and elite genotypes	Metabolome	Flag leaf	Ion chromatography-mass spectrometry, High-performance liquid chromatography	Identified mQTLs for metabolites linked with antioxidant defense	[43]
*Solanum lycopersicum*	Introgression lines	Secondary metabolites	Fruit	Ultra performance liquid chromatography	Identified 679 mQTLs for secondary metabolites	[45]
*Solanum lycopersicum*	Introgression lines	Secondary metabolites	Fruit	Ultraperformance liquid chromatography-tandem mass spectrometry	Identified mQTLs which decrease the variability for primary and secondary metabolites called canalization metabolite quantitative trait loci (cmQTL)	[46]
*Solanum lycopersicum*	Introgression lines	Metabolome	Fruit	Gas chromatography–mass spectrometry	Identified putative 30 mQTLs for amino acids and organic acids	[27]
*Solanum lycopersicum*	RIL	Metabolome	Germinating seed	Gas chromatography-time-of-flight/mass spectrometry	Identified mQTLs for metabolites within several QTL hotspots	[48]
*Brassica napus*	Tapidor × Ningyou7 (DH)	Glucosinolates	Leaf and seed	High-performance liquid chromatography	Identified 105 mQTLs that affected glucosinolate concentration in either or both of the organs	[44]
*Oryza sativa*	Landraces and elite varieties	Metabolome	Grains	Liquid chromatography electrospray ionization tandem mass spectrometric	Identified new candidate genes which influence important metabolic and/or morphological traits	[16]
*Oryza sativa*	Landraces accessions	Secondary metabolites	Leaf	Liquid chromatography quadrupole time-of-flight mass spectrometry	Identified 323 associations among 143 SNPs and 89 metabolites	[54]
*Oryza sativa*	Landraces accessions	Phenolamides	Leaf	Liquid chromatography–mass spectrometry	Identified two spermidine hydroxyl cinnamoyl transferases (Os12g27220 and Os12g27254) that could underline the natural variation levels of spermidine conjugates in rice	[51]
*Oryza sativa*	Landraces accessions	Metabolome	Leaf	Liquid chromatography–mass spectrometry	Identified 36 candidate genes controlling metabolite levels which are of potential physiological and nutritional significance	[34]
*Zea mays*	Inbred lines	Metabolome	Leaf	Gas chromatography–mass spectrometry	Identified 26 distinct metabolites with potential associations with SNPs, explaining up to 32.0% of genetic variance	[41]
*Zea mays*	Inbred lines	Oil components	Kernel	Ultra-performance liquid chromatography	Reported 74 loci potentially associated with kernel oil concentration and fatty acid content	[57]
*Zea mays*	Inbred lines	Tocochromanol	Grain	High-performance liquid chromatography	Identified favorable *ZmVTE4* haplotype and three novel gene targets for increasing the level of vitamin E and antioxidant	[58]
*Zea mays*	Inbred lines	Carotenoid	Grain	High-performance liquid chromatography	Identified 58 candidate genes involved in carotenoids biogenesis and retention in maize	[59]
*Zea mays*	Inbred lines	Metabolome	Kernel	Liquid chromatography–mass spectrometry	Identified significant causal variants for five candidate genes associated with metabolic traits	[50]
*Triticum aestivum*	Elite lines	Metabolome	Flag leaf	Gas chromatography–mass spectrometry	Reported potential associations for 6 metabolic characters, namely oxalic acid, ornithine, L-arginine, pentose alcohol III, L-tyrosine, and a sugar oligomer (oligo II), with between 1 and 17 associated SNPs	[55]
*Solanum lycopersicum* L.	Landrace accessions	Metabolome	Fruit	Gas chromatography–mass spectrometry	Identified 44 loci linked with 19 traits, including sucrose, ascorbate, malate, and citrate levels	[60]

**Table 2 cells-10-00346-t002:** Metabolic engineering towards enhancing performance of plants.

	Gene	Function of Gene	Phenotypes of Transgenics	Reference
Phytohormones Engineering to Enhance Abiotic Stress Tolerance				
ABA	*LOS5*	Key regulator of ABA biosynthesis	Enhanced ABA accumulation and drought tolerance in maize	[83]
	*AtLOS5*		Enhanced salinity tolerance attributed to enhanced Na^+^ efflux and H^+^ influx	[84]
	*MsZEP*	Vital role in ABA biosynthesis	Heterologous expression of gene resulted in better salt and drought tolerance	[85]
	*SnRK2*.*4*	Protein kinase involved in ABA signaling and root architecture maintenance	Exhibited enhanced tolerance to abiotic stress and improved photosynthesis in *Arabidopsis*	[86]
Auxin	*YUCCA6*	Auxin/IPA biosynthesis gene	Overexpression enhanced tolerance to drought and oxidative stress	[87]
	*OsIAA6*	Auxin/IAA gene family member	Enhanced drought tolerance via auxin biosynthesis regulation in transgenic rice	[88]
	*IPT*	Controls rate limiting step of cytokinin biosynthesis	Transgenic tomato showed enhanced growth and yield under salt stress	[89]
Cytokinin	*CKX*	Cytokinin dehydrogenase	Overexpression led to enhanced drought tolerance in transgenic *Arabidopsis*	[90]
	*AtCKX1*		Overexpression led to enhanced drought tolerance through dehydration avoidance in transgenic barley	[91]
	*ERF-1* (*JERF1*)	Response factors of ethylene and jasmonates	Enhanced drought tolerance in rice	[92]
Ethylene	*ACC-Synthase*	Catalyzes rate-limiting step in ethylene biosynthesis	Transgenic maize showed reduced ethylene levels with better drought tolerance (gene silencing)	[93]
	*ZmARGOS*	Negative regulators of ethylene signal transduction	Enhanced drought tolerance in transgenic *Arabidopsis* and maize	[94]
	*OsGSK1*	BR negative regulator	Improved tolerance of knockout mutants to cold, heat, salt, and drought stresses	[95]
Brassinosteroids	*AtHSD1*	Role in BR biosynthesis	Overproduction enhanced growth, yield, and salinity tolerance	[96]
	*BdBRI1*	BR-receptor gene	Down-regulation improved drought tolerance with dwarf phenotypes of purple false brome	[97]
Metabolic Engineering of Secondary Metabolic Pathways Genes				
Flavonoid Biosynthetic Pathway	*MYB12*	Transcription factor, regulate the biosynthesis of phenylpropanoid	Overexpression in *Arabidopsis* enhanced drought and salt tolerance	[98]
	*DFR-OX B*	Catalyzes the reduction of dihydroflavonols to leucoanthocyanidins in anthocyanin biosynthesis	Overexpression in *Brassica napus* enhanced drought and salt tolerance	[99]
	*PFG1/PAP1*		Overexpression in *Arabidopsis* enhanced oxidative and drought tolerance	[100]
Carotenoid Biosynthetic Pathway	*β-LCY1*	Involved in *beta*-carotene biosynthesis pathway	Overexpression in *Nicotiana tabacum* enhanced drought and salt tolerance	[101]
			Inhibition in *Arabidopsis* and *Nicotiana* enhanced salinity tolerance	[102]
IPP biosynthetic pathway	*GGPS*	Involved in the synthesis of an osmolyte glucosyl glycerol	Overexpression in *Arabidopsis thaliana* enhanced osmotic stress tolerance	[103]
Metabolic Engineering for Enhancing Photosynthetic Efficiency				
Light Harvesting Enzyme	*PsbS*	Plays a crucial role in xanthophyll-dependent nonphotochemical quenching	Overexpression increases leaf CO_2_ uptake and plant dry matter productivity in tobacco	[104]
			Overexpression reduces water loss per CO_2_ assimilated in tobacco	[105]
Calvin–Benson cycle	*SBPase*	Key regulator of carbon flux	Overexpression enhances photosynthesis against high temperature stress in transgenic rice	[106]
			Overexpression increases photosynthetic carbon assimilation, leaf area, and biomass yield in tobacco	[107]
			Overexpression increases photosynthesis and grain yield in wheat	[108]
Photorespiration	*GCS H-protein*	Catalyzes the degradation of glycine	Overexpressing increases biomass yield in transgenic tobacco plants	[109]
	*GDC-L protein*	Catalyzes the tetrahydrofolate-dependent catabolism of glycine	Overexpression increased rates of CO_2_ assimilation, photorespiration, and dry weight in *Arabidopsis*	[110]
	*GDC-T protein*	Tetrahydrofolate dependent protein, catalyzes glycine	Overexpression neither altered photosynthetic CO_2_ uptake nor plant growth in *Arabidopsis*	[111]
Electron Transport	Algal Cyt c6	Participates in algal photosynthetic electron transport chain	Overexpression increase CO_2_ assimilation rates and plant growth in *Arabidopsis*	[112]
			Constitutive expression enhanced water use efficiency, chlorophyll and carotenoid content in tobacco	[113]
	Rieske FeS	Regulates electron transfer	Constitutive expression enhanced photosynthetic electron transport rates, chlorophyll and carotenoid content	[114]
Carbon transport	Cyanobacterial inorganic carbon transporter B	Regulates CO_2_ concentration mechanism	Significantly higher photosynthetic rates and biomass was observed in overexpressed Arabidopsis lines	[115,116]
			Overexpression enhanced CO_2_ assimilation rates in rice and tobacco	[117]
Genome Editing Mediated Metabolic Engineering				
CRISPR/Cas9 multiplex gene editing	*IFS (isoflavone synthase)*	Plays significant role in biosynthesis of isoflavonoids	Mutation enhanced isoflavone content and resistance to soya bean mosaic virus (SMV)	[118]
	*GmSPL9 genes*	Regulate plant architecture	Targeted mutagenesis altered plant architecture and yield in soybean	[119]
	*SGR (Stay green)*	Regulates plant chlorophyll degradation and senescence	Significantly improved lycopene content in tomato fruit	[120]
	*SAPK2*	Primary mediator of ABA signaling	Enhanced sensitivity to drought stress and ROS in rice	[121]
	*ARGOS8*	Negative regulator of ethylene responses	Enhanced drought tolerance and yield in maize	[122]
	*SIMAPK3*	Participates in SA or JA defense-signaling pathways	Enhanced drought tolerance in tomato	[123]
Metabolic Engineering for Biofortification of Vitamin A, Fe and Zn				
Vitamin A	*Phytoene synthase (PSY) and phytoene desaturase(CrtI)* gene	Participate in carotenoid biosynthetic pathway	Enhanced nutritional value of golden rice by increasing provitamin A content	[124]
			Increase Increase total carotenoid content in transgenic wheat	[125]
Iron (Fe)	*Soyfer H-*1	Soybean ferritin gene involved in storage of iron	Overexpression enhanced iron content in rice seed	[126]
	*OsNAS2*	Participates in iron-acquisition	Overexpression enhanced Fe and Zn content in rice endosperm	[127]
Zinc (Zn)	*HvNAS1* (*Nicotianamine Synthas*)	Metal chelator, involved in accumulation of Fe and Zn	Overexpressing enhanced Fe and Zn contents in the leaves, flowers, and seeds in rice	[128]
Metabolic Engineering for Abiotic Stress Tolerance				
Transcription Factor	*TTG2*	WRKY TF regulates diverse biological processes	Regulate trichome development and enhance salinity tolerance in *Brassica*	[129]
	*ERF-2 (like)*	Ethylene response TF, regulates various stress responses	Overexpression enhanced submergence tolerance in *Arabidopsis*	[130]
	*NAC 19, 82*	TF plays important roles in development, abiotic, biotic stress responses, and biosynthesis	Overexpression led to regulate ROS and cell death in tobacco leaves	[131]
	*HSFA4A*	Heat shock transcription factor	Enhanced desiccation tolerance in seeds and activate antioxidant system in *Arabidopsis*	[132]
	*CDF1*	Regulates expression of floral activator genes	Regulate flowering time and freezing tolerance in *Arabidopsis*	[133]
Kinases	*MAPKKK 4*	Regulates growth, development, and immune responses	Regulation of ROS induced cell death in tobacco leaves, lipid peroxidation, and DNA degradation	[134]
	*MAPKKK 18, 19*	Regulates plant immunity and hormone responses	Regulates ROS formation and cell death in tobacco	[135]
	*CPK2*	Regulates cellular responses to various stimuli	Regulates ROS and cell death control through interaction with RbohD in tobacco	[136]
	*MKK1*	Regulates stresses, growth, and development	Enhanced response of plants to pathogenic bacteria and drought stress in tobacco	[137]
Transporters	*SWEET*	Plays important role in sucrose translocation and crop yields	Regulates plant growth and development and also participates in biotic and abiotic stress response	[138]
	*HMA*	Heavy metal ATPase, response to Cd stress	Played an important role in Cd translocation in the leaves of *Brassica napus*	[139]
	*ABC*	Regulates uptake and allocation of metabolites and xenobiotics	Significantly induced under Cd stress and regulate ion channels	[140]
	*AQPs* *(Aquaporins)*	Facilitates molecule movement across the membranes	Overexpression enhances salt stress tolerance in transgenic tobacco	[141]
Metabolic Engineering for Terpenoids/Volatile Compounds				
Monoterpenoids	*Linalool synthase (LIS)*	Catalyzes the formation of acyclic monoterpene linalool	Transgenic petunia plants result in the accumulation of S-linalyl-beta-D-glucopyranoside	[142]
			Engineering of terpenoid pathway led enhanced aroma and flavor in tomato	[143]
	*Limonene Synthase*	Catalyzes the cyclization of geranyl pyrophosphate to (4S)-limonene	Modified essential oil content in transgenic lines in transgenic mint	[144]
	*β*-*Glucosidase*	Catalyzes the hydrolysis of the glycosidic bonds and release glucose	Affects the emission of plant volatiles, plant-environment communication and aroma	[145]
Sesquiterpenoids	*Trichodiene synthase*	Catalyzes the formation of trichodiene	Transgenic tobacco enhanced the expression of active enzyme and low-level accumulation of its sesquiterpenoid product	[146]
	zingiberene synthase (ZIS)	Catalyzes the reaction forming zingiberene and other mono- and sesquiterpenes	Overexpression led to enhanced both mono-and sesquiterpene content in tomato fruit	[147]
	*Germacrene A synthase*	Key cytosolic enzyme of sesquiterpene lactone biosynthesis pathway	Transgenic lines with strong transgene expression showed growth retardation and *FaNES1*-expressing lines enhanced the resistance against the aphids	[148]
Diterpenoids	*Taxadiene synthase*	Catalyzes the chemical reaction geranylgeranyl diphosphate	Enhanced level of toxoids was found in genetically engineering plant	[149]
Metabolic Engineering for Biotic Stress Tolerance				
Pathogen Perception	*EFR* (EF-Tu *receptor*)	Pattern recognition receptor (PRR), binds to prokaryotic protein EF-TU	Expression in susceptible genotypes reduced bacterial wilt incidence and enhanced yield	[150]
	*Bs2*	Bs2 gene is a member of the NBS-LRR class of R genes	Transgenic tomato conferred resistance to bacterial spot disease	[151]
Pathogen Effector Binding	*Os11N3/OsSWEET14*	Encode sucrose transporters	Transgenic wheat provided effective resistance to *Fusarium graminearum*	[152]
	*Xa27*	Important R-*genes*, effective against *Xoo*	Provided resistance to different strains of *Xoo* and bacterial leaf streak	[153]
Defence Signaling Pathways	NPR1	Master immune regulatory gene	Mediate broad-spectrum disease resistance without compromising plant fitness in *Arabidopsis thaliana* and rice	[154]
	IPA1/OsSPL14	Regulate rice plant architecture	Enhanced yield and disease resistance in rice	[155]
Recessive Resistance Alleles	*Mlo (Mildew Locus O)*	Knockdown resulted in powdery mildew resistance	Loss of function mutation confer resistance to powdery mildew fungi	[156]
	*bs5*	Recessive genes resistant to bacterial spot	Confers disease resistance against *Xanthomonas euvesicatoria* in pepper and tomato	[157]
Dominant Resistance Proteins	*PFLP*	Ferrodoxin like protein, involved in redox reactions	Overexpression induced hypersensitive reaction and resistance in tobacco	[158]
	*Lr34*	Wheat multipathogen resistant gene	Confer resistance to anthracnose and rust in sorghum	[159]
	*Oxalate oxidase*	Participates in degradation of oxalic acid	Enhanced resistance to *Sclerotinia sclerotium* in oilseed rape	[160]
Antimicrobial Compound Production	Rs-AFP defensin *(**Raphanus sativus* antifungal *protein)*	Antifungal plant *defensins*	Transgenic wheat conferred resistance to *Fusarium graminearum* and *Rhizoctonia cerealis*	[161]
	Virus KP4	Fungal killer toxin encoded by RNA virus	Transgenic wheat showed resistance to loose smut	[162]
	*MsrA1*	Involved in mannan biosynthesis	Transgenic *Brassica Juncea* exhibited resistance to fungal phytopathogens	[163]
RNAi Mediated	*AC1* from bean golden mosaic virus	Modulates virus induced gene silencing	Transgenic common bean (*Phaseolus vulgaris*) conferred resistance to ban golden mosaic virus	[164]
	*Coat protein* gene from potato virus Y	Protects RNA genome	Exhibited resistance to mixed virus infection in potato	[165]

**Table 3 cells-10-00346-t003:** List of the pro-vitamin-A biofortified crops.

Crops	Genes with Donor Organism	Carotenoid Content	References
Rice	*Narcissus pseudonarcissus* (*crtB*)	Combination of transgenes enabled biosynthesis of provitamin A in the rice endosperm (Golden Rice 1)	[171]
	*Erwinia uredovora* (*crtI*)		
	*Zea mays* (*PSY*)	Increase in total carotenoids up to 23-fold (Golden Rice II)	[124]
	*Erwinia uredovora* (*crtI*)		
Wheat	*Zea mays* (*PSY*)	The total carotenoids content was increased up to 10-fold	[125]
	*Erwinia uredovora* (*crtI*)		
	*Erwinia uredovora* (*crtB*, *crtI*)	Total carotenoid content increased by 8-fold and beta-carotene content increased by 65-fold	[181]
	*Erwinia uredovora* (*crtB*)	Increase in the beta-carotene content by 31-fold	[182]
	*Triticum aestivum* (*HYD*)		
Potato	*Pantoea ananatis* (*crtB*)	Total carotenoid increased by 4-fold with major increase in beta-carotene and lutein content	[183]
	*Pantoea ananatis* (*crtE*)	Total carotenoid up by 2.5-fold and beta-carotene content by 14-fold	[184]
	*Pantoea ananatis* (*crtB*, *crtI*, *crtY*)	Total carotenoid increased by 20-fold and that of beta-carotene by 3600-fold	[185]
	*Solanum tuberosum* (*β-CHX*)	Beta-carotene content was increased from trace level to 3.31 μg/g FW	[186]
	*Brassica oleracea* (*Or*)	Carotenoid content was increased by 10-fold	[177]
Corn	*Zea mays* (*PSY*)	Increased level of beta-carotene content including hydroxy- and keto-carotenoids	[187]
	*Gentiana lutea* (*LCYE*, *β-CHX*)		
	*Paracoccus* (*crtW*)		
	*Pantoea ananatis* (*crtI*)		
	*Pantoea ananatis* (*crtB*, *crtI*, *zds*)	Total carotenoids up by 34-fold with preferential accumulation of beta-carotene	[188]
	*Zea mays* (*PSY*)	The transgenic kernels contained 169-fold the normal amount of β-carotene	[189]
	*Pantoea ananatis* (*crtI*)		
Tomato	*Erwinia uredovora* (*crtI*)	The β-carotene content increased about threefold, up to 45% of the total carotenoid content	[190]
	*Solanum lycopersicum* (*LCYB*)	7-fold increase in fruit beta-carotene content	[172]
	*Arabidopsis thaliana* (*LCYB*)	12-fold increase in beta-carotene content along with beta-cryptoxanthin and zeaxanthin accumulation	[191]
	*Capsicum annuum* (*β-CHX*)		
	*Erwinia uredovora* (*crtB*)	Total fruit carotenoids upby 2–4-fold in fruits	[192]
	*Solanum lycopersicum* (*LCYB*)	Carotenoid content was increased by 2-fold while beta-carotene is up by 27-fold	[173]
	*Arabidopsis thaliana* (*HMGR*)	Total carotenoid content increased by 1.6-fold and beta-carotene by 2.2-fold	[193]
	*Escherichia coli* (*dxs*)		
	*Capsicum annuum* (*FIB*)	Total carotenoid content was up by 2-fold	[194]
	*Narcissus pseudonarcissus* (*crtY*)	4.5-fold increase in beta-carotene and >50% increase in total carotenoid accumulation	[195]
	*Citrus* (*LCYB1*)	Beta-carotene level was increased by 4.1-fold, and the total carotenoid content increased by 30% in the fruits	[196]
Cassava	*Erwinia uredovora* (*crtB*)	Total carotenoidcontent increase by 15-fold and that of beta-carotene by 37-fold	[197]
	*Arabidopsis thaliana* (*DXS*)		
	*Phytoene synthase*	Total carotenoid content increased by 33-fold and beta-carotene by 15-fold	[198]
	*Bacterial* (*crtB*)	Total carotenoid content increased by 30-fold with beta-carotene accounting for 80–90% of total carotenoid content	[199]
	*Arabidopsis thaliana* (*DXS*)		
Sorghum	*Zea mays (PSY)*	24-fold increase in beta-carotene content	[200]
	*Pantoea ananatis* (*crtI*)		
	*Arabidopsis thaliana* (*DXS*)		
	*Hordeum vulgare* (*HGGT*)		
Melon	*Or*	Total carotenoid content increased by 11-fold	[176]
Cauliflower	*Or*	Beta-carotene content increased by 7-fold	[201]

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
