# Peer review of "Metabolomics Intervention Towards Better Understanding of Plant Traits"

_cells, 2021, doi:10.3390/cells10020346_

Round 1

Reviewer 1 Report

The paper requires extensive review for English grammar, style, and often word choice. Sentences are often incomplete and not comprehensible. The review of metabolomic studies that has enabled mapping of mQTLs and similar is sound, the summaries of crop improvements seems disconnected to the topic of metabolomics however. A large section is dedicated to the engineering of nitogen fixation in non-nitrogen fixing species, but appears to have little to do with genomics.

Author Response

Response to the comments of the Reviewer#1

  1. The paper requires extensive review for English grammar, style, and often word choice. Sentences are often incomplete and not comprehensible.

Authors’ Response: Many thanks for this valuable suggestion. We have now improved it significantly.

  1. The review of metabolomics studies that has enabled mapping of mQTLs and similar is sound, the summaries of crop improvements seem disconnected to the topic of metabolomics however. A large section is dedicated to the engineering of nitrogen fixation in non-nitrogen fixing species, but appears to have little to do with genomics.

Authors’ Response: Thanks for the feedback. We have modified MS as per the Reviewers’ suggestions. We have divided “Nitrogen fixation in non-nitrogen fixing species” into two sections and made appropriate changes.

Reviewer 2 Report

The authors have proposed a review on the topic: Metabolomics Intervention Towards Better Trait 2 Understanding in Plants

The review is well-documented, gives a good view of existing work, although being highly descriptive and maybe would deserve a bit of development on some topics such as What is the impact of metabolic engineering on crop phenotype? This might be the subject of a short paragraph, as well as on the public acceptance/ stakeholders policies.

Lots of problems with writing as spaces are missing in a lot of sentences, probably due to the saving in pdf format, but beside this, grammar should also be revised as some sentences are difficult to understand as such. Therefore, the reviewer recommends to thoroughly revise the language before resubmitting a new version of the manuscript.

Table 2: In general, difficult to read. The links between the lines are not always easy to follow. Why some sentences are underlined?

In table 3, not always obvious the link between the crop and the genes targeted, the beta-caroten content and the genetic approach, please prepare a table that is clear. The beta-caroten content, what does this represent in terms of improvement of the content?

Please revise the paragraph 3.3. e.g. the lines on pinus radiate (?) are not finished obviously; lines 68 to 75, the plant is not indicated.

Please also note that Crisper-cas9 should be Crispr-cas9

Due to these remarks, the reviewer recommends a major revision.

Author Response

Response to the comments of the Reviewer#2

  1. The authors have proposed a review on the topic: Metabolomics Intervention Towards Better Trait 2 Understanding in Plants

The review is well-documented, gives a good view of existing work, although being highly descriptive and maybe would deserve a bit of development on some topics such as What is the impact of metabolic engineering on crop phenotype? This might be the subject of a short paragraph, as well as on the public acceptance/ stakeholders policies.

Authors’ Response: Thank you for kind appreciation of our work. We are grateful to you for providing valuable suggestion and important comments. At many places the role of metabolic engineering and phenotypes has been discussed in this MS, for instance discussion about carotenoids and pro-Vitamin A enhancement in different crop, etc. The section 3 highlights similar observation. Therefore, we haven’t added any extra section related to impact of metabolic engineering on crop phenotype, because other reviewer has expressed to reduce the length of there MS. We hope review will agree to our point and we are thanking him in advance.

Additionally, as per the reviewer suggestion, we have added separate section Public perception for the metabolic engineered plants (section 6) in the MS.

  1. Lots of problems with writing as spaces are missing in a lot of sentences, probably due to the saving in pdf format, but beside this, grammar should also be revised as some sentences are difficult to understand as such. Therefore, the reviewer recommends to thoroughly revise the language before resubmitting a new version of the manuscript.

Authors’ Response: Thanks for the suggestions, we have significantly improved the newer version of MS.

  1. Table 2: In general, difficult to read. The links between the lines are not always easy to follow. Why some sentences are underlined?

Authors’ Response: We have updated the Table 2.

In table 3, not always obvious the link between the crop and the genes targeted, the beta-caroten content and the genetic approach, please prepare a table that is clear. The beta-caroten content, what does this represent in terms of improvement of the content?

Authors’ Response: Thanks for raising this point, we have updated Table 3.

  1. Please revise the paragraph 3.3. e.g. the lines on Pinus radiate (?) are not finished obviously; lines 68 to 75, the plant is not indicated.

Authors’ Response: Many thanks for this suggestion, we have corrected it now.

  1. Please also note that Crisper-cas9 should be Crispr-cas9

Authors’ Response: Thanks for pinpointing this typo error, we have replaced Crisper-cas9with Crispr-cas9